



# Inter-comparison and evaluation of sea ice type concentration algorithms

Yufang Ye[1], Mohammed Shokr[2], Signe Aaboe[3], Wiebke Aldenhoff[1], Leif E.B. Eriksson[1], Georg Heygster[4], Christian Melsheimer[4], and Fanny Girard-Ardhuin[5]

[1]Department of Space, Earth and Environment, Chalmers University of Technology, Gothenburg, Sweden
[2]Meteorological Research Division, Environment and Climate Change Canada, Toronto, Canada
[3]Department of Remote Sensing and Data Management, Norwegian Meteorological Institute, Tromso, Norway
[4]Institute of Environmental Physics, University of Bremen, Bremen, Germany
[5]Ifremer, University of Brest, CNRS, IRD, Laboratoire d'Oceanographie Physique et Spatiale (LOPS), IUEM, Brest, France

**Correspondence:** Yufang Ye (yufang@chalmers.se)

**Abstract.** Sea ice has been monitored in terms of concentration and types with microwave satellite observations since the late 1970s. However, it remains an open question as to which sea ice type concentration (SITC) method is most appropriate for ice type distribution and hence climate monitoring. This paper presents key results of inter-comparison and evaluation for eight SITC methods. The SITC methods were inter-compared with two sea ice age (SIA) and three sea ice type (SIT) products
using microwave radiometer and scatterometer data from 2000 to 2015. Their performances were evaluated quantitatively with samples that are used for generating tie points, and qualitatively with the RADARSAT imagery. The methods that combined scatterometer and radiometer data have overall better performances on ice type discrimination. The best methods are ECICE-QSCAT-2 for the years 2000-2009 and ECICE-ASCAT for 2009-2015, both using scatterometer data along with radiometer data. Although the SIA and SIT products are fairly good datasets for delineating ice type distributions, the SITC methods are
better on preserving details like varied concentration of different ice types and work better under specific sea ice conditions, for instance, homogeneous sea ice regions with little artifact for SIA algorithms to track.

## 1 Introduction

Sea ice is an important component of the global climate system and at the same time a sensitive indicator of climate change. Its
area has decreased dramatically in the Arctic over the past four decades. Moreover, the Arctic sea ice is increasingly dominated by thinner and younger first-year ice (FYI) instead of multiyear ice (MYI), the ice that has survived at least one summer melt. It has been reported that the ratio of MYI to total sea ice area in the Arctic has been shrinking sharply during the last decades, from over two thirds in the mid 1980s to less than one third in 2017 (Maslanik et al., 2011; Kwok, 2018). As the MYI area in the central Arctic declines, the associated mean sea ice thickness and volume decreases significantly (Kwok, 2004; Polyakov





et al., 2012; Kwok and Cunningham, 2015). Kwok (2018) found that the MYI area anomalies can explain about 85% of the variance in the anomalies in Arctic sea ice volume. In the context of sea ice volume, the shift towards the Arctic ice cover dominated by FYI is significant since it changes the expected seasonal cycle of mass balance. Consequently, it impacts the weather and climate in the Arctic and sub-Arctic regions through different thermodynamic and dynamic processes (Vihma, 2014; Overland et al., 2015; Petrie et al., 2015).

Sea ice has been monitored in terms of concentration and types with microwave remote sensing observations since the late 1970s. Total sea ice concentration (ice/water discrimination) can be derived reliably from microwave data due to distinct microwave signatures between open water and sea ice. Several algorithms have been developed to estimate total sea ice concentrations from microwave radiometer and scatterometer data. A partial list is provided in (Ivanova et al., 2014, 2015), with comparison of their performance and evaluation over low and high sea ice concentrations. Among them, only a few algorithms

are able to distinguish different ice types and estimate their partial concentrations since it poses much more difficulties, mainly due to the lower dynamic range in the brightness temperatures of FYI and MYI (Spreen et al., 2008). The partial ice concentration methods include 1) those who discriminate between two types of ice, e.g., between FYI and MYI (Cavalieri et al., 1984; Comiso et al., 1997; Kwok, 2004; Comiso, 2012; Ye and Heygster, 2015), between thin ice and ice with snow layering (Markus and Cavalieri, 2000); 2) and the one that retrieves concentration of any given set of ice types based on input of probability distri-

bution of each remote sensing observation for each ice (surface) type (Shokr et al., 2008). Due to the availability of few in-situ data, the exact validation of these sea ice type concentration (SITC) methods is difficult. All the above-mentioned methods are validated indirectly, using scattered in-situ information, operational sea ice maps, or limited ship observations. However, the question remains as to which SITC retrieval method is most appropriate for Arctic sea ice type distribution estimates hence climate monitoring.

In this study we use ice type concentration retrievals from eight selected SITC methods to investigate the differences between their results and hence assess their performances. Sea ice age (SIA) and Sea ice type (SIT) products (i.e. ice type classification products) were also used for inter-comparison and assessment of their performances. The study used microwave satellite data from winter months (October-April) of 2000-2015. This paper is organized as follows. Section 2 describes the data. It includes microwave radiometer and scatterometer data, which are used in the SITC methods, as well as SIA and SIT products and

ancillary data sets. Section 3 introduces the SITC algorithms and the samples for generating tie points. Section 4 presents the main results of this work: comparison of the SITC results against SIA and SIT products, and evaluation against samples used to generate tie points, then against the RADARSAT imagery. Conclusions are highlighted in Section 5.

## 2 Data

### 2.1 Microwave radiometer and scatterometer data

Passive and active microwave satellite data are used in monitoring sea ice because of their ability of working in the absence of sunlight and through clouds. Passive microwave sensors, i.e., radiometers, measure the emitted radiation from Earth's surface (in terms of brightness temperature). Active microwave sensors, e.g., scatterometers, measure the backscattered radar signal





after reflection off the surface (in terms of backscatter coefficient). In this study, radiometer and scatterometer data were used in the eight SITC methods, the description of which can be found in Section 3. The retrieved SITC results were thereafter
inter-compared with the SIA and SIT products.

The radiometer data used in this study includes data from the Special Sensor Microwave Imager (SSM/I) and the Special Sensor Microwave Imager/Sounder (SSMIS) aboard the Defense Meteorological Satellite Program. The swath data from January 2000 to December 2015 is taken from the Fundamental Climate Data Record (FCDR) of Microwave Image Radiance, Edition 3, provided by the European Organisation for the Exploitation of Meteorological Satellites (EUMETSAT) Climate
Monitoring Satellite Application Facility (Fennig et al., 2017). The radiometer ratio parameters to be used in the algorithms (see Equation 1) were computed on swath data before the brightness temperatures and the respective parameters were gridded to the Equal-Area Scalable Earth (EASE2) Grid of 25 km spcaing.

The scatterometer data used are from the SeaWinds scatterometer onboard QuikSCAT (QSCAT) satellite and the Advanced Scatterometer (ASCAT) on EUMETSAT's Metop-A and Metop-B satellites. QSCAT is a Ku-band (13.4 GHz) conically scan-
ning pencil-beam scatterometer, operating from July 1999 to November 2009. It has two beams, each with a wide range of azimuth angles. The inner beam is horizontally polarized at an incidence angle of $46°$, whereas the outer beam is vertically polarized at an incidence angle of $54°$ (Hoffman and Leidner, 2005). Normalized backscatter coefficients of horizontal and vertical polarizations, $\sigma_{hh}^{\circ}$ and $\sigma_{vv}^{\circ}$ (referred to as BKSH and BKSV, respectively), are resampled to a grid of 12.5 km in polar stereographic projection. They are provided and archived at CERSAT/Ifremer (Girard-Ardhuin et al., 2008), available on
CERSAT/Ifremer portal (http://cersat.ifremer.fr). ASCAT is a C-band scatterometer with three vertically polarized antennas transmitting pulses at 5.255 GHz, operating from May 2007 to present. The fan-beam antennas are oriented at $45°$, $90°$ and $135°$ with respect to the satellite track. The incidence angle varies between $25°$ and $65°$. Backscatter coefficients from ASCAT ($\sigma_a^{\circ}$, referred to as BKAS) are normalised to the backscatter at a constant incidence angle of $40°$ and resampled to a polar stereographic 12.5 km resolution grid. The gridded data is built from swath data (from EUMETSAT Level 2B product) averaged
over one day by CERSAT/Ifremer. The QSCAT data from January 2000 to November 2009 and ASCAT data from October 2007 to December 2015 (data from Metop-A only) were used for sea ice type concentration retrieval in this study.

## 2.2 Sea ice age products

Two sea ice age (SIA) products were used in this study, one from the National Snow and Ice Data Center (referred to as NSIDC-SIA) and the other developed by Korosov et al. (2018) for the Sea Ice Climate Change Initiative (SICCI) project of
the European Space Agency (referred to as SICCI-SIA product). Both products are based on the concept of backtracking ice trajectories however differ on techniques.

The weekly NSIDC-SIA product is available in an EASE grid with 12.5 km spacing for the period from January 1984 to December 2018 (Tschudi et al., 2019). It is generated from positions of virtual Lagrangian ice parcels (Fowler et al., 2004), where the grid cell is assigned the age of the oldest ice parcel. This means that the assigned ice age is not necessarily the one
that has the largest fraction. The dominant drifters within a cell could be younger ice types, while the entire cell is still assigned to old ice.



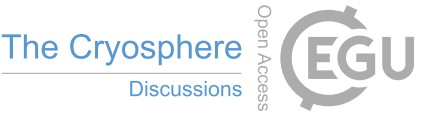

The SICCI-SIA product was generated on a polar stereographic grid with 20 km spacing, available from October 2012 to September 2017. The product is based on an Eulerian advection scheme, which accounts for observed divergence/convergence and freezing/melting of sea ice (Korosov et al., 2018). The algorithm uses daily sea ice drift and sea ice concentration products

from the Ocean and Sea Ice Satellite Application Facility (OSISAF) (Lavergne, 2016a, b; Tonboe et al., 2017). Here the grid cell is not assigned single ice age but fractions of different ice age. The main advantage of this product is the ability to generate the fraction of individual ice age in each pixel, which makes it more comparable to SITC results.

## 2.3   Sea ice type products

The sea ice type (SIT) products used in this study include one from the Copernicus Climate Change Service (referred to as C3S-

SIT) and another one developed by Rivas et al. (2018) from the Royal Netherlands Meteorological Institute (KNMI) (referred to as KNMI-SIT). The C3S-SIT product uses microwave radiometer data only, whereas KNMI-SIT is a purely scatterometer-based product.

In the C3S-SIT product, the sea ice types (MYI or FYI, both having ice concentration >30%) are assigned from atmospherically corrected brightness temperatures using a Bayesian approach (Aaboe et al., 2019) that computes the probability

of occurrence of the most likely ice type. For the period from 2000 to 2015, the radiometer data consists of the FCDR of Microwave Imager Radiances, described in Section 2.1. The C3S-SIT dataset is provided for the winter months (October to April) on a 25 km EASE grid, available through the C3S Climate Data Store (https://cds.climate.copernicus.eu).

The KNMI-SIT product uses a maximum likelihood class discrimination approach (Rivas et al., 2018) based on probabilistic distance to ocean wind and sea ice geophysical model functions (GMFs). GMFs describe the behaviour of backscatter as a

function of observational geometry (i.e., incidence and azimuth angles) and geophysical variables such as wind speed and direction or sea ice type. This dataset is available in a polar stereographic 12.5 km grid for the whole period of satellite scatterometer missions (ERS, QSCAT and ASCAT), extending from 1992 to present. Two KNMI-SIT products were used in the study, one based on QSCAT data (referred to as KNMI-QSCAT-SIT product, from 2000 to 2009), the other with ASCAT data (referred to as KNMI-ASCAT-SIT product, from 2007 to 2015).

## 2.4   Other data

Four RADARSAT-1 and two RADARSAT-2 synthetic aperture radar (SAR) images were used for validation. RADARSAT-1 and RADARSAT-2 are two Canadian Earth Observation Satellites. The first one was operational from March 1995 to March 2013 and the second from December 2007 to present. RADARSAT-2 is a follow-on satellite mission of RADARSAT-1, both with SAR instrument working at 5.3 GHz (C-band). All the SAR images used in this study are ScanSAR Mode data, with a

pixel size of 50 m and resolution of around 100 m. They were radiometrically calibrated and projected to the UTM projection with a pixel size of 100 m. The geographic locations of the SAR images and the dates are shown in Figure 1. The data was acquired from the Alaska Satellite Facility (ASF) and RADARSAT-2 portal in MacDonald Dettwiler and Associates (MDA).



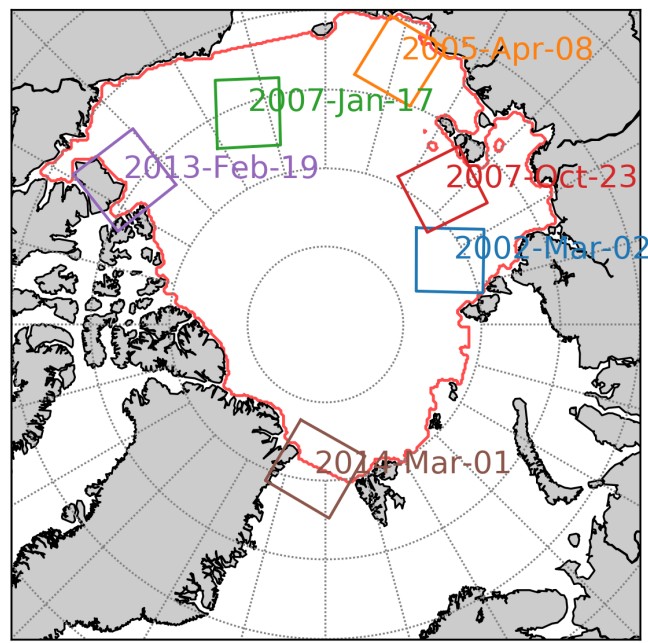

**Figure 1.** Geographic locations of the SAR images and outline of the Arctic Basin (red contour, provided by (Rivas et al., 2018)).

## 3 Methods

### 3.1 Three sea ice type concentration algorithms

In this study, we investigated the following three algorithms for sea ice type concentration retrievals: the NASA Team (NT) algorithm (Cavalieri et al., 1984; Steffen and Schweiger, 1991), the Bootstrap (BT) algorithm (Comiso, 2012) and the ECICE algorithm (Shokr et al., 2008). NT and BT are purely radiometer-based algorithms, while ECICE allows combination of radiometer and scatterometer observations to estimate the concentration of any number of given ice types as long as it is less than the number of observations.

The BT algorithm was first presented in Comiso (1990) for sea ice type concentrations using vertically polarized brightness temperatures at 19 GHz and 37 GHz ($TB19V$ and $TB37V$). In Comiso (2012), the authors made an improvement to the algorithm by using dynamic tie points. Instead of using brightness temperatures directly, the NT algorithm uses polarization ratio (PR) and gradient ratio (GR) to calculate the concentration of FYI and MYI. These two independent ratios are:

$$PR19 = (TB19V - TB19H)/(TB19V + TB19H)$$

$$GR37V19V = (TB37V - TB19V)/(TB37V + TB19V),$$

(1)

where $TB19H$ is the horizontally polarized brightness temperature from the 19 GHz channel and other parameters are defined in the same manner.





The Environment Canada's Ice Concentration Extractor (ECICE) algorithm starts with a linear mixing model that decomposes each observation into contributions from each surface type (in our case FYI, MYI and OW) weighted by their concentrations. This algorithm requires a priori probability distribution function for each observation (input parameter) from each of

the given surface types. The number of input parameters must be equal to or larger than the number of surface types. In this study, six sets of input parameters were used in ECICE: two sets of purely radiometer data, and four sets of combined data (scatterometer and radiometer). Each set is comprised of four input parameters. The six sets of input parameters used in ECICE along with those in the NT and BT algorithms are presented in Table 1.

**Table 1.** Input parameters and the resolution of the sea ice type concentration methods.

| Method | Input parameters | Resolution (km) |
|---|---|---|
| BT | $TB19V, TB37V$ | 25 |
| NT | $PR19, GR37V19V$ | 25 |
| ECICE-SSMI-1 | $TB19H, TB19V, TB37V, GR37V19V$ | 25 |
| ECICE-SSMI-2 | $TB19V, TB37H, TB37V, GR37V19V$ | 25 |
| ECICE-QSCAT-1 | $BKSH, BKSV, GR37V19V, PR19$ | 12.5 |
| ECICE-QSCAT-2 | $BKSH, BKSV, TB37H, TB37V$ | 12.5 |
| ECICE-ASCAT-1 | $BKAS, TB37H, GR37V19V, PR19$ | 12.5 |
| ECICE-ASCAT-2 | $BKAS, TB37H, TB37V, GR37V19V$ | 12.5 |

## 3.2    Tie points and samples

To develop tie points for the NT and BT algorithms as well as probability distributions of the input parameters for ECICE (all for FYI, MYI and OW), samples were taken during the overlapping period of ASCAT and QSCAT, in the years of 2007-2009. OW samples were taken randomly in the climatological open water area, which is 1) outside the annual sea ice maximum extent; 2) north of 60°N; and 3) outside the land mask that extends 100 km into open sea. FYI and MYI samples were manually selected based on inspection of time series of QSCAT backscatter maps at 4.45 km resolution, which are generated by Brigham

Young University using the technique presented in Early and Long (2001), and atmospheric temperature maps. MYI samples usually have higher backscatter and are mainly located north of the Canadian Archipelago, while FYI samples usually have lower backscatter and extend outward from the core of the Arctic Basin. In total, 15686, 6493 and 4894 samples were selected from the three-year data of OW, FYI and MYI, respectively. Spatial and temporal distributions of the samples for the three surface types of the year 2007 are as shown in Figure 2.





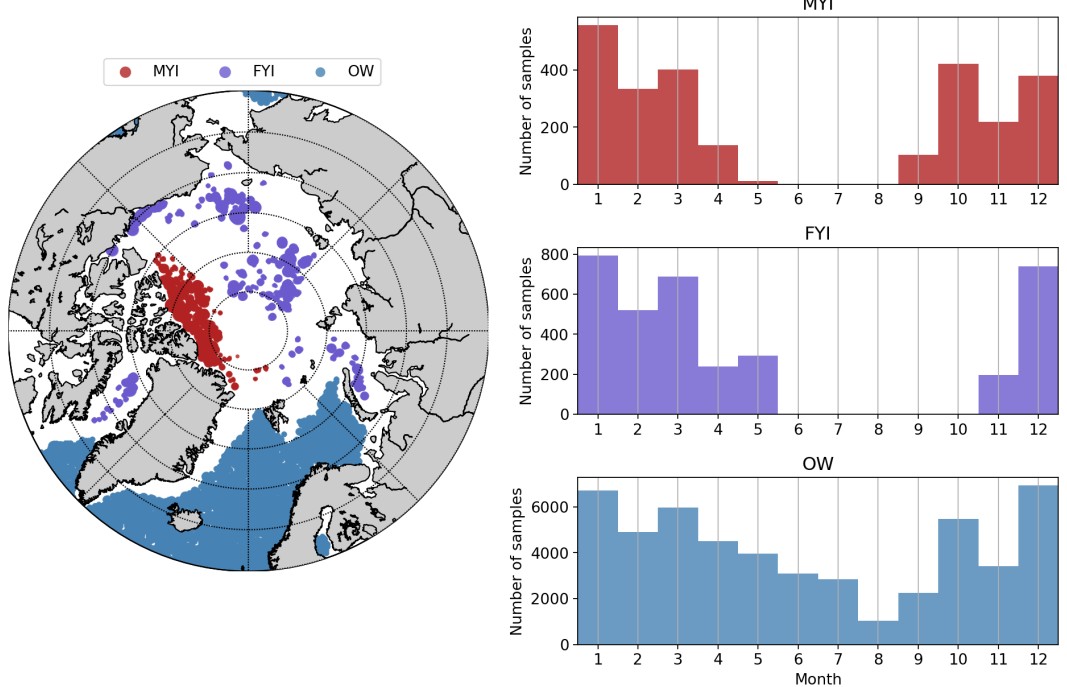

**Figure 2.** Spatial (left) and monthly (right) distributions of the MYI, FYI and OW samples for the year 2007.

Averages of these samples were regarded as tie points in the BT and NT algorithms, whereas their probability distribution functions (PDF) were used in ECICE for fair comparison. The PDFs and tie points for the brightness temperatures of all samples are shown in Figure 3.

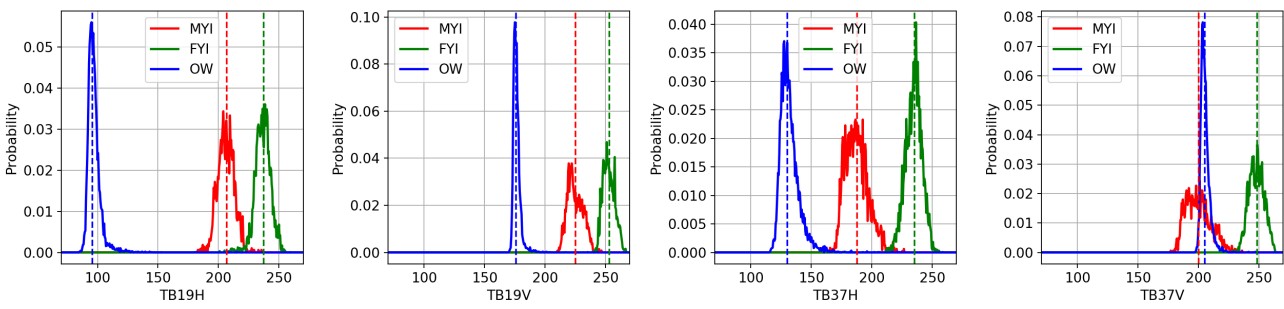

**Figure 3.** Probability distribution functions of the brightness temperatures from all the samples (used in ECICE) and the corresponding tie points (vertical dashed lines, used in BT and NT).



## 4  Results

This section starts with a comparison between the two SIA products, NSDIC-SIA and SICCI-SIA. It then proceeds with
comparison between the results from the eight SITC methods and each SIA product, and afterwards against the three SIT
products. While the comparison provides clues for the best SITC algorithm, the follow-up evaluation against tie points and the
SAR images provides more concrete proof. All the comparisons of this study were performed on the data of a pre-defined area
within the Arctic Basin and limited by the polar hole of 87.8°N. Outline of the pre-defined Arctic Basin is as delineated by the
red contour in Figure 1, provided by Rivas et al. (2018). In this area, MYI is the dominant ice type and ambiguities between
MYI and deformed FYI are negligible compared to marginal ice zone.

### 4.1  Comparison with sea ice age products

### 4.1.1  NSIDC SIA product vs SICCI SIA product

The area of a given ice age from the NSIDC-SIA product is estimated as the integral of the area of all pixels that contain the
corresponding age. On the other hand, the area from the SICCI-SIA product is estimated as the integral of the fractions for the
given ice age. This feature of SICCI-SIA product (the ability to generate the fraction of individual ice age in each pixel) makes
it more suitable for comparison against results from the SITC methods.

Figure 4 shows a time series of MYI area estimates from the NSIDC-SIA and SICCI-SIA products for the freezing seasons
of 2012 throughout 2017 in the Arctic Basin. In the two sub-figures, MYI area from six SITC methods are superimposed.
MYI area from the NSIDC-SIA product is overall higher than the SICCI-SIA product, which is expected from the way of
calculating the area. For the winters of 2012-2017, the average MYI area from NSIDC-SIA and SICCI-SIA is $2.53 \times 10^9$ km$^2$
and $1.89 \times 10^9$ km$^2$, respectively. For the winter of 2012-2013, the MYI is only represented by second-year ice (SYI, no older
ice) in the SICCI product since calculations were initiated in that winter and all MYI was assigned to be SYI. As calculations
proceed, the percentage of SYI decreases in following winters and reaches 53.5% in the winter of 2016-2017, which is very
close to that of NSIDC-SIA (57.6%). Within each winter, the MYI area has similar declining patterns from both products. The
average reduction between October and April is $916.81 \times 10^3$ km$^2$ and $713.54 \times 10^3$ km$^2$ from the SICCI and NSIDC product,
respectively.





**Figure 4.** Area of sea ice age from NSIDC-SIA (a) and SICCI-SIA (b) products and MYI area from six SITC methods from October 2012 to April 2017.

### 4.1.2 SITC results versus SIA products

Comparison between the daily MYI area from the SITC methods against SIA products from NSDIC and SICCI is depicted in Figure 4. In the ideal case, the SITC data should match the upper green boundary. It can be seen that MYI area from all the SITC algorithms are in better agreement with the SICCI-SIA than NSDIC-SIA product. The SITC methods produce smaller MYI area compared to the NSDIC product, especially in the beginning of the freezing season (October and November).



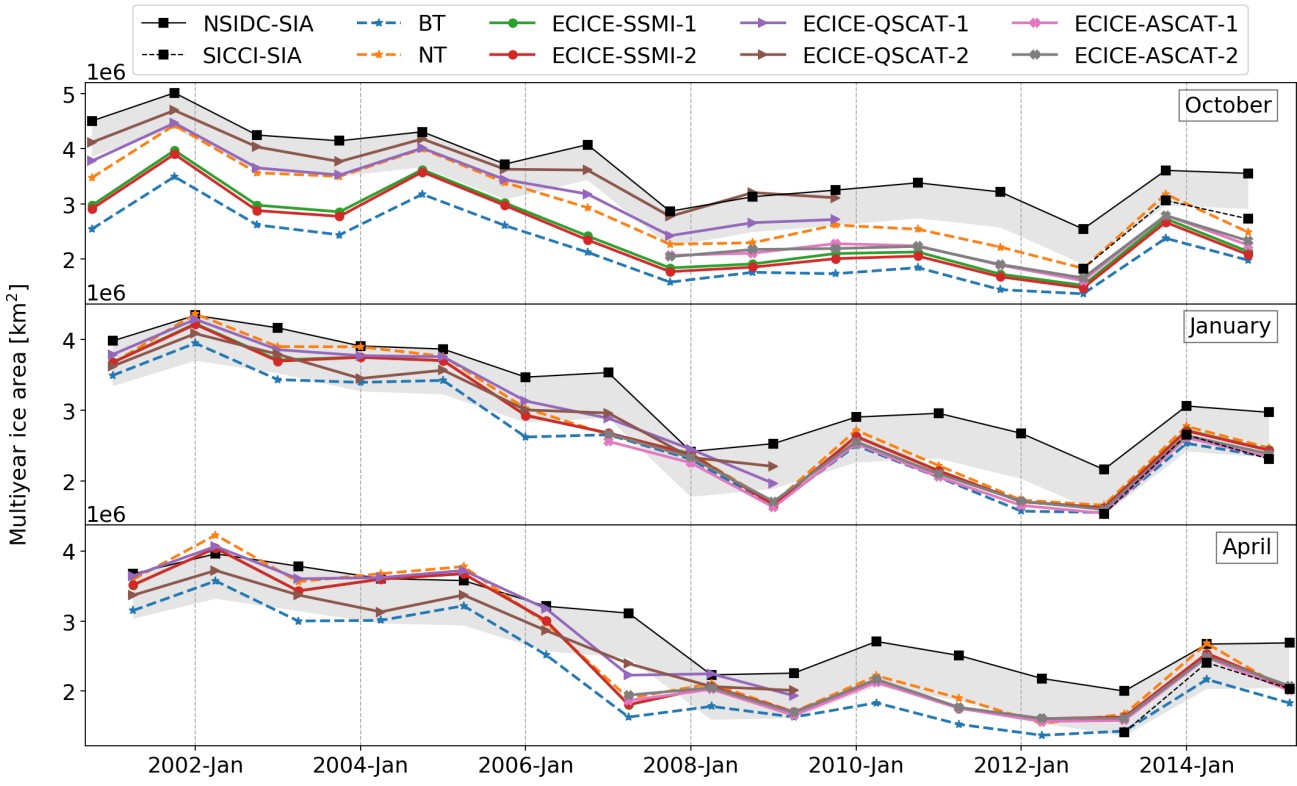

**Figure 5.** Monthly average MYI area between all the SITC and SIA products in October (top pannel), January (middle) and April (bottom) from October 2000 to April 2015. The grey shaded area represents the average MYI area difference between NSIDC-SIA and SICCI-SIA.

**Table 2.** Mean differences (MD) and mean absolute differences (MAD) of MYI area between 6 SITC products and the adjusted value from NSIDC-SIA for January, October, and April of the years 2000-2009.

| Method | Oct. - MD | Oct. - MAD | Jan. - MD | Jan. - MAD | Apr. - MD | Apr. - MAD |
|---|---|---|---|---|---|---|
| | ($\times 10^3$ km$^2$) | | ($\times 10^3$ km$^2$) | | ($\times 10^3$ km$^2$) | |
| BT | -874.78 | 874.78 | **54.22** | **225.43** | **-15.39** | **215.56** |
| NT | **-42.31** | **206.80** | 323.77 | 420.09 | 430.32 | 557.85 |
| ECICE-SSMI-1 | -502.22 | 502.22 | 267.03 | 359.43 | 366.62 | 512.51 |
| ECICE-SSMI-2 | -568.48 | 568.48 | 262.21 | 352.83 | 368.21 | 512.93 |
| ECICE-QSCAT-1 | **100.58** | **175.28** | 384.09 | 385.04 | 511.00 | 565.45 |
| ECICE-QSCAT-2 | 419.31 | 419.31 | **287.16** | **287.16** | 294.61 | 312.45 |



**Table 3.** Mean differences (MD) and mean absolute differences (MAD) of MYI area between 6 SITC products and the adjusted value from NSIDC-SIA for January, October, and April of the years 2007-2015.

| Method | Oct. - MD | Oct. - MAD | Jan. - MD | Jan. - MAD | Apr. - MD | Apr. - MAD |
|---|---|---|---|---|---|---|
| | ($\times 10^3$ km$^2$) | | ($\times 10^3$ km$^2$) | | ($\times 10^3$ km$^2$) | |
| BT | -789.13 | 789.13 | **-3.68** | **236.10** | **-68.91** | **225.49** |
| NT | **-122.03** | **188.27** | 137.52 | 287.39 | 211.73 | 225.49 |
| ECICE-SSMI-1 | -525.58 | 525.58 | 109.55 | 279.79 | 182.80 | 207.44 |
| ECICE-SSMI-2 | -589.29 | 589.29 | 105.22 | 275.33 | 186.35 | 209.89 |
| ECICE-ASCAT-1 | -396.76 | 396.76 | **100.55** | **233.61** | **134.94** | **172.79** |
| ECICE-ASCAT-2 | -386.41 | 386.41 | **61.91** | **239.79** | 168.93 | 193.62 |

As mentioned in Section 2.2, the backtracking technique used in the SICCI-SIA products makes it a better source for SITC methods comparison. However, due to the better availability, the NSIDC-SIA product was used for long-term comparison instead of the SICCI-SIA product. Comparison of MYI area between SITC and SIA product is presented in the form of monthly averages as shown in Figure 5. As indicated in Section 4.1.1, MYI area from the NSIDC-SIA product is consistently larger than that from the SICCI-SIA product, by 641.36 $\times 10^3$ km$^2$ on average. This value is subtracted from the MYI area from the NSIDC-SIA product and this new parameter is referred to as "adjusted MYI area from NSIDC-SIA". It is represented by the lower boundary of the shaded area in Figure 5. Mean differences (MD) and mean absolute differences (MAD) between the SITC MYI area and the NSIDC-SIA adjusted MYI area are listed in Tables 2 and 3 for the QSCAT and ASCAT period, respectively. The MD represents any potential bias of the SITC MYI area relative to the NSIDC-SIA adjusted MYI area, whereas the MAD represents the magnitude of the mean spreading from the NSIDC-SIA adjusted MYI area. The MAD can be large even when there is no bias.

For October, in the top panel of Figure 5, all the SITC methods show less MYI than the NSIDC-SIA product, while the MYI area from the NT method exhibits a good agreement with the SICCI-SIA product. MYI area from the ECICE-QSCAT-1, ECICE-QSCAT-2 and NT method fall either completely or mostly into the shaded area of the NSIDC-SIA time series, which indicates that these three methods agree well with the SIA products in October. For October throughout the years of 2000-2015, the ECICE-QSCAT-1 product has the smallest MAD of MYI area among all the SITC methods and on average gives approximately 1 $\times 10^5$ km$^2$ higher estimate of MYI area than the NSIDC-SIA adjusted MYI (Tables 2 and 3). On the other hand, the NT method gives the smallest bias in both periods (MD of -42.31 $\times 10^3$ km$^2$ and -122.03 $\times 10^3$ km$^2$ in Tables 2 and 3, respectively) but slightly larger MAD than the ECICE-QSCAT-1 product (Table 2). For the period of ASCAT (2007-2015), the ECICE-ASCAT methods are the second best.

For January, in the middle panel of Figure 5, the spread of the MYI area from the eight SITC methods is the smallest compared to October and April. This is expected from the cold temperatures and stable physical properties of sea ice in January, which leads to smaller uncertainties than beginning and end of winter. Most of the SITC MYI area fall between





the values from NSIDC-SIA and SICCI-SIA. In the QSCAT period (2000-2009), the BT method yields the smallest MYI area difference, and the ECICE-QSCAT-2 has the second lowest (see Table 2). In the ASCAT period (2007-2015), the three smallest MYI area differences are those from the ECICE-ASCAT-1, BT and ECICE-ASCAT-2 methods (Tabel 3).

In April, in the bottom panel of Figure 5, the NT, ECICE-QSCAT-1 and ECICE-SSMI-2 methods produce more MYI than the NSIDC-SIA product in some years, which could be a potential overestimation of MYI and can be confirmed in Section 4.4

(the case in the East Siberian Sea). Although the BT method yields the lowest MD in April throughout the years of 2000-2015 (see Table 2 and 3), it produces less MYI than any other SITC and SIA product in other months, which is regarded as a sign of underestimation for MYI. Except for BT, the smallest MYI area difference is that from ECICE-QSCAT-2 for the QSCAT period and ECICE-ASCAT-1 for the ASCAT period.

## 4.2 Comparison with sea ice type products

Sea ice type (SIT) and SITC products are widely used in climate studies. However there has been no study focusing on the comparison of them. This subsection provides an overall comparison of the monthly MYI concentrations from the SITC methods for the three sets of ice pixels that are identified in each SIT product. For the C3S-SIT product, the three sets of pixels are those of FYI, MYI and MIX (pixels of either FYI or MYI). For the KNMI-SIT product, the three sets are FYI, SYI and MYI, where SYI represents the yonger multiyear ice and MYI here represents multiyear ice older than two years. Results from

the SITC methods are considered to match the SIT product if the MYI concentration is high for MYI (and SYI) pixels from the latter and low for FYI pixels.

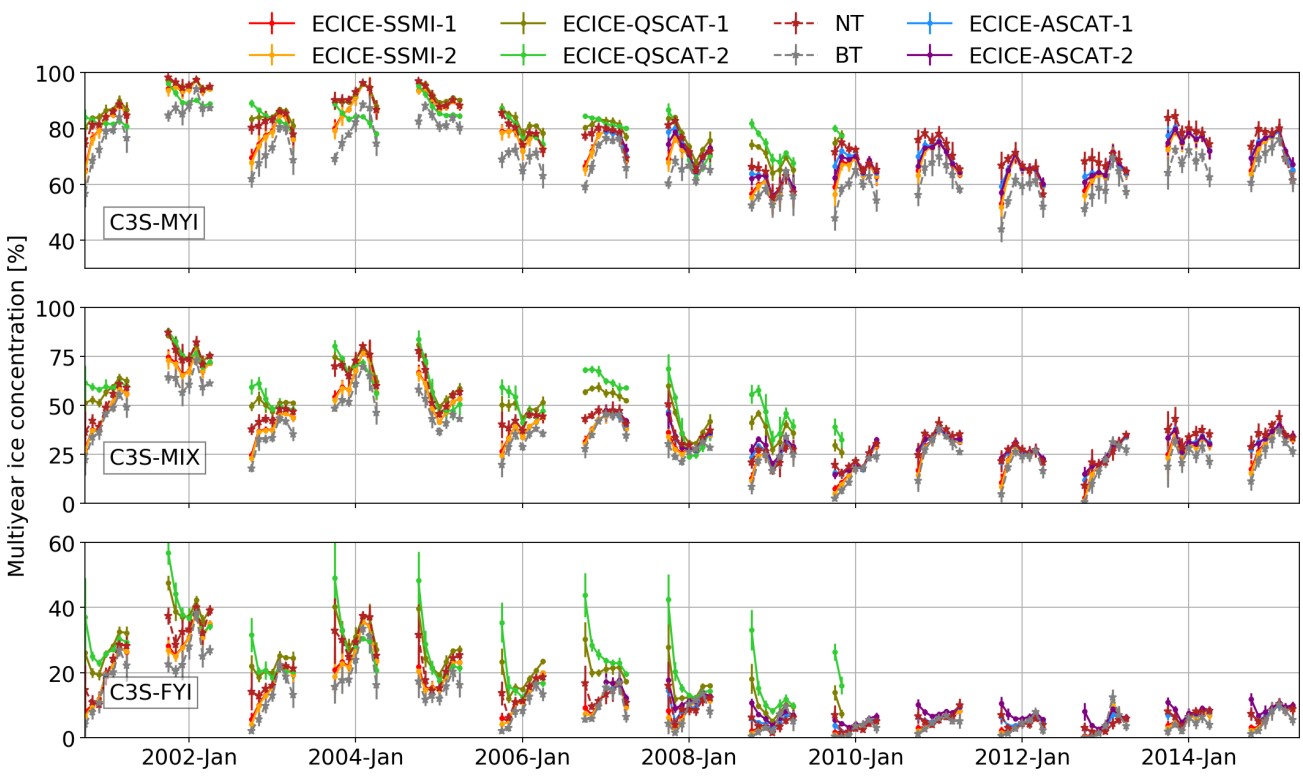

**Figure 6.** Monthly average MYI concentrations from SITC methods over FYI, MIX and MYI pixels in the C3S-SIT product.





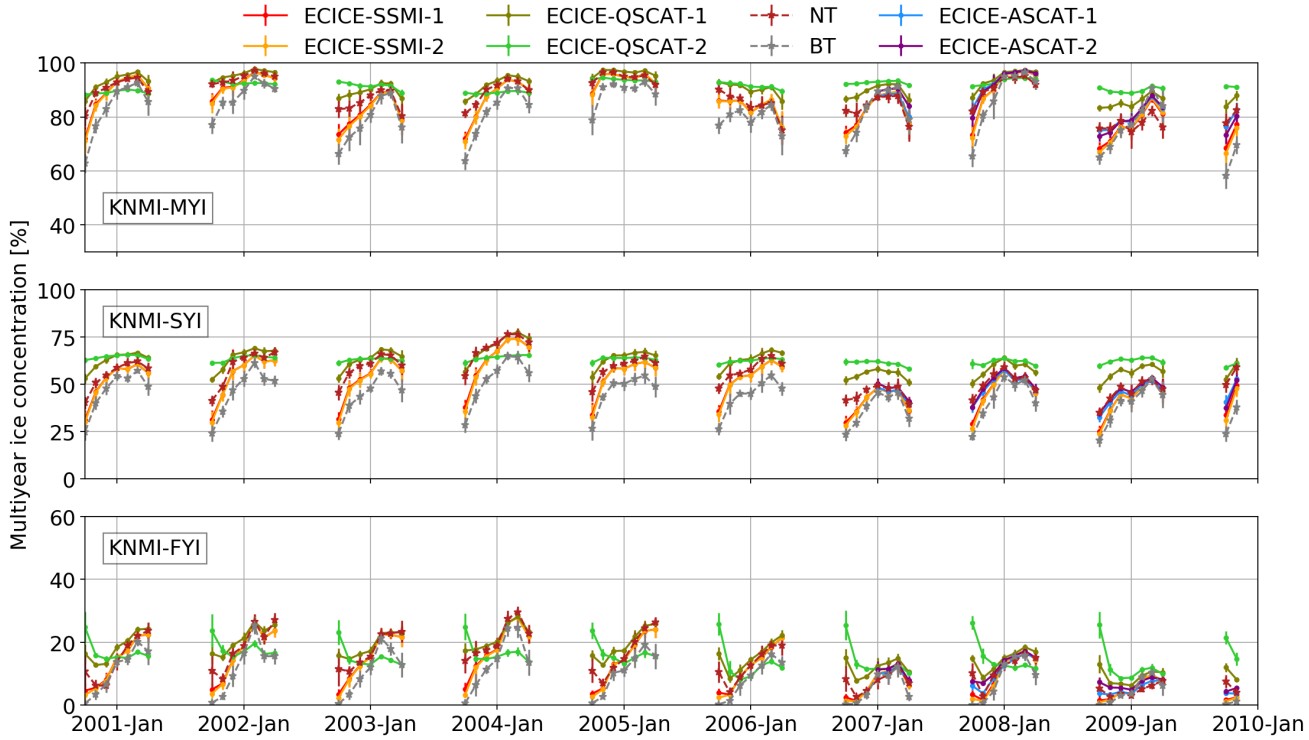

**Figure 7.** Monthly average MYI concentrations from SITC methods over FYI, SYI and MYI pixels in the KNMI-QSCAT-SIT product.





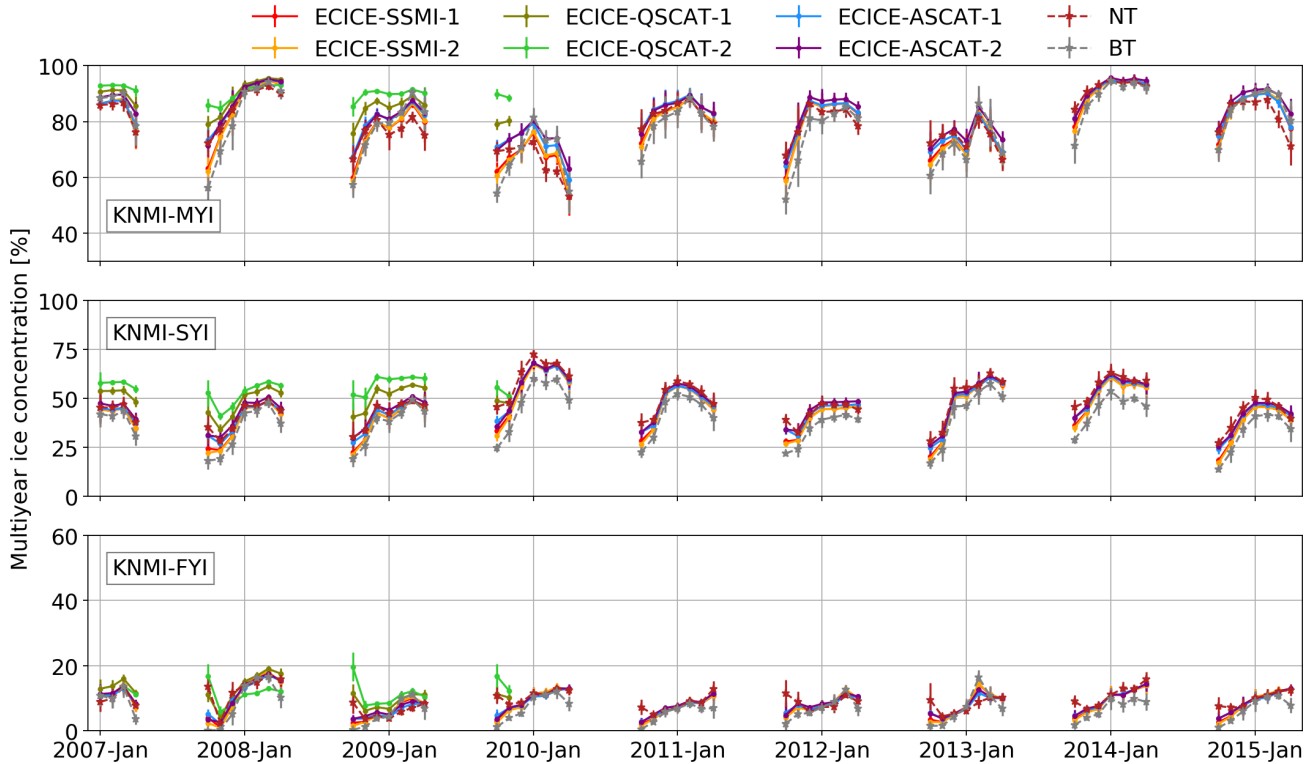

**Figure 8.** Monthly average MYI concentrations from the SITC methods over FYI, SYI and MYI pixels in the KNMI-ASCAT-SIT product.

Comparison with the SITC products (Figures 6, 7 and 8) shows that MYI pixels identified in the three SIT products have MYI concentrations mostly above 60% from all the SITC methods except BT. The BT method exhibits clear underestimation of MYI concentration. While the overall MYI concentration for MYI pixels identified in the C3S-SIT product is around 60%-100%,

that of the KNMI-QSCAT-SIT and KNMI-ASCAT-SIT is around 80%-100% and 70%-90%, respectively. In the beginning and end of the winters, there is an apparent underestimation of MYI concentrations from all the SITC methods except the ECICE-QSCAT methods. The overall higher MYI concentration from the QSCAT-based SITC product (ECICE-QSCAT) could be explained by the higher sensitivity of Ku-band scatterometer to volume scattering in MYI (Ezraty and Cavanié, 1999).

MYI concentrations for the FYI pixels in the KNMI-SIT products (Figures 7 and 8) from all the SITC methods are well below

30%, while large values are found in the FYI pixels of the C3S-SIT product (see Figures 6). The large MYI concentrations (up to 60%) are found in October and November of the years 2000-2009, especially from the ECICE-QSCAT methods. This indicates that the FYI pixels in the C3S-SIT product may contain large fraction of MYI, especially in the QSCAT period, which is a sign of misidentification. For such cases, it is more the anomalous changes on brightness temperatures and QSCAT backscatters in the beginning of winter, e.g. due to increasing snow wetness (Ye et al., 2016a), than the algorithm that lead to

the misidentification of the SITC and C3S-SIT product.





The MIX pixels in the C3S-SIT product are those where the C3S-SIT algorithm has problems to distinguish between FYI and MYI. Therefore a large spread of the MYI concentrations from different SITC methods for these pixels are expected (see middle panel in Figure 6). The SYI pixels of the KNMI-SIT products correspond to the youngest multiyear ice, which might still capture some of the typical features that are used to identify FYI. Within each winter, MYI concentrations for the MIX

(in C3S-SIT product) and SYI pixels (in KNMI-SIT products) have large variations and similar changing trends as those for the MYI pixels (see middle panel in Figures 6, 7 and 8). During the winters from 2000 to 2015, there is no clear pattern for the MYI concentrations of the C3S-SIT MIX pixels, whereas those for the SYI pixels of the KNMI-SIT products mostly have clear increasing trend from October to January and stabilize to a value of 50%-70% until April.

The above discussions highlight two observations. Firstly, ECICE-QSCAT appears to be the most suitable approach to

reproduce the information in the SIT products. However, the use of QSCAT data in ECICE may lead to exaggerated MYI concentrations in the beginning of winter (October-November), especially for FYI pixels of the C3S-SIT product. Secondly, the convex shape of the average MYI concentration for the MYI pixels of SIT products in each winter indicates that all the SITC methods agree more with the SIT products during cold dry winter conditions and less during transition seasons.

### 4.3 Validation with samples/tie points

This section presents a quantitative validation using samples for tie points that were used in the SITC methods, whereas the next section gives a qualitative validation using SAR images. The validation against tie points is to describe the overall performance of the SITC methods, while the latter validation is to show their performances under specific conditions.

As described in Section 3.2, averages of OW, FYI and MYI samples are regarded as tie points in the NT and BT methods, while PDFs of the corresponding input parameters from the same samples are used in the ECICE algorithm to generate desired

sets of tie points. In this subsection, we use satellite observations from these samples and derive the concentration for the corresponding surface type in the sampled areas. For instance, MYI concentrations are retrieved from the SITC methods for all the MYI samples, while FYI concentrations are retrieved for all the FYI samples. The same is performed on the OW samples. For perfect performance, the SITC method should return 100% for the samples of the same surface type and 0% for samples of other types. Table 4 shows the average values of OW, FYI and MYI concentrations from the eight SITC methods. Histograms

of the output concentrations of FYI and MYI for the relevant samples are presented in Figure 9.





**Table 4.** Average OW, FYI and MYI concentrations of the OW, FYI and MYI samples, respectively.

| Method | OW (%) | FYI (%) | MYI (%) |
|---|---|---|---|
| BT | 98.28 | **93.84** | 92.23 |
| NT | 98.36 | **92.66** | 91.65 |
| ECICE-SSMI-1 | 98.56 | 90.94 | 92.08 |
| ECICE-SSMI-2 | 98.57 | 91.15 | 92.01 |
| ECICE-QSCAT-1 | **99.81** | 90.81 | **95.99** |
| ECICE-QSCAT-2 | **99.81** | **94.05** | **97.92** |
| ECICE-ASCAT-1 | 98.27 | 91.46 | 94.12 |
| ECICE-ASCAT-2 | **99.45** | 90.74 | 94.23 |

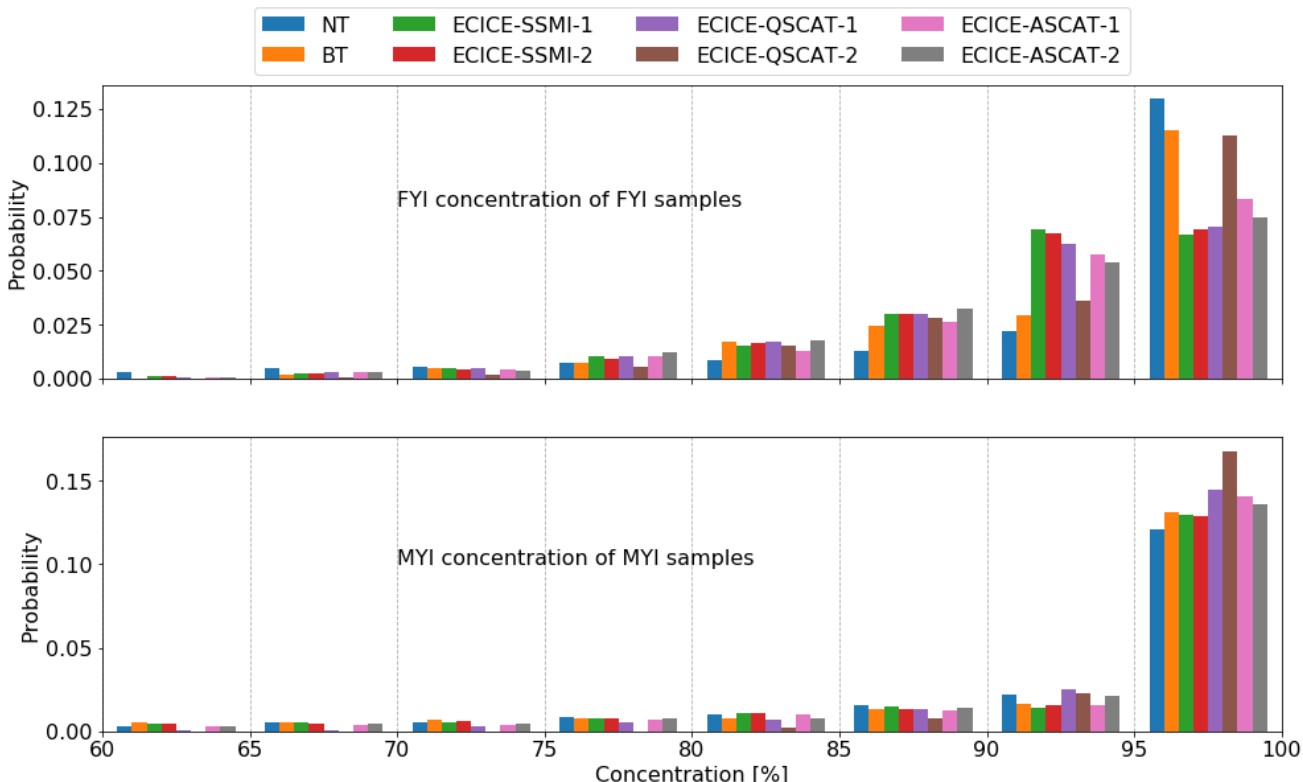

**Figure 9.** Histograms of their concentrations for the FYI and MYI samples.

For the OW samples, all the SITC methods identify OW quite accurately, with concentrations all above 98%. This is due to the fact that the microwave signatures from water and ice are much more distinct than those for different ice types. The three





highest OW concentrations (all above 99%) are those from ECICE-QSCAT-1, ECICE-QSCAT-2 and ECICE-ASCAT-2. As for the FYI samples, the three largest FYI concentrations are obtained from ECICE-QSCAT-2 (94.05%), BT (93.84%) and NT 265 (92.66%). In addition, histograms of the FYI concentrations show wide distributions from all the SITC methods. The overall lower average concentrations and wide distributions indicate that all the SITC methods listed here tend to misidentify FYI.

For MYI samples, the average MYI concentration differs more among the SITC methods than that for FYI. It reaches 97.92% for ECICE-QSCAT-2 and 95.99% for ECICE-QSCAT-1, while it is only 91.65% for NT. The four scatterometer-based methods have overall higher MYI concentrations than the other algorithms, which can be served as a proof for the good performance of 270 using scatterometer data along with radiometer data for sea ice type detection. Besides, MYI concentrations from all the SITC methods are mainly limited to values between 95% and 100%. It reveals that MYI can be identified fairly accurately by all the SITC methods most of the time. Misidentification of MYI could occur under certain weather conditions, examples of which will be shown in Section 4.4.

### 4.4 Validation with SAR

This section presents qualitative comparisons of the MYI concentrations from different SITC methods against information retrieved from visual analysis of the RADARSAT-1 and RADARSAT-2 SAR images. Six case studies are addressed using four RADARSAT-1 and two RADARSAT-2 images (HH polarization channel only) to present cases under different sea ice conditions. The geographic locations of the images are shown in Figure 1. In each case, maps of MYI concentration from examined SITC algorithms and SICCI-SIA are presented along with the sea ice age map from NSIDC-SIA and weather 280 information from ECMWF ERA5 reanalysis (air temperature and surface wind field from 12:00 UTC). In all the SITC maps, OW has been masked out and is shown as white color in Figures 10-15. Maps of ECICE-ASCAT-1 and ECICE-SSMI-2 are sometimes not shown in the figures since they are usually quite similar to those of ECICE-ASCAT-2 and ECICE-SSMI-1, respectively.



### 4.4.1 Case studies (1) and (2): ice in the Laptev Sea

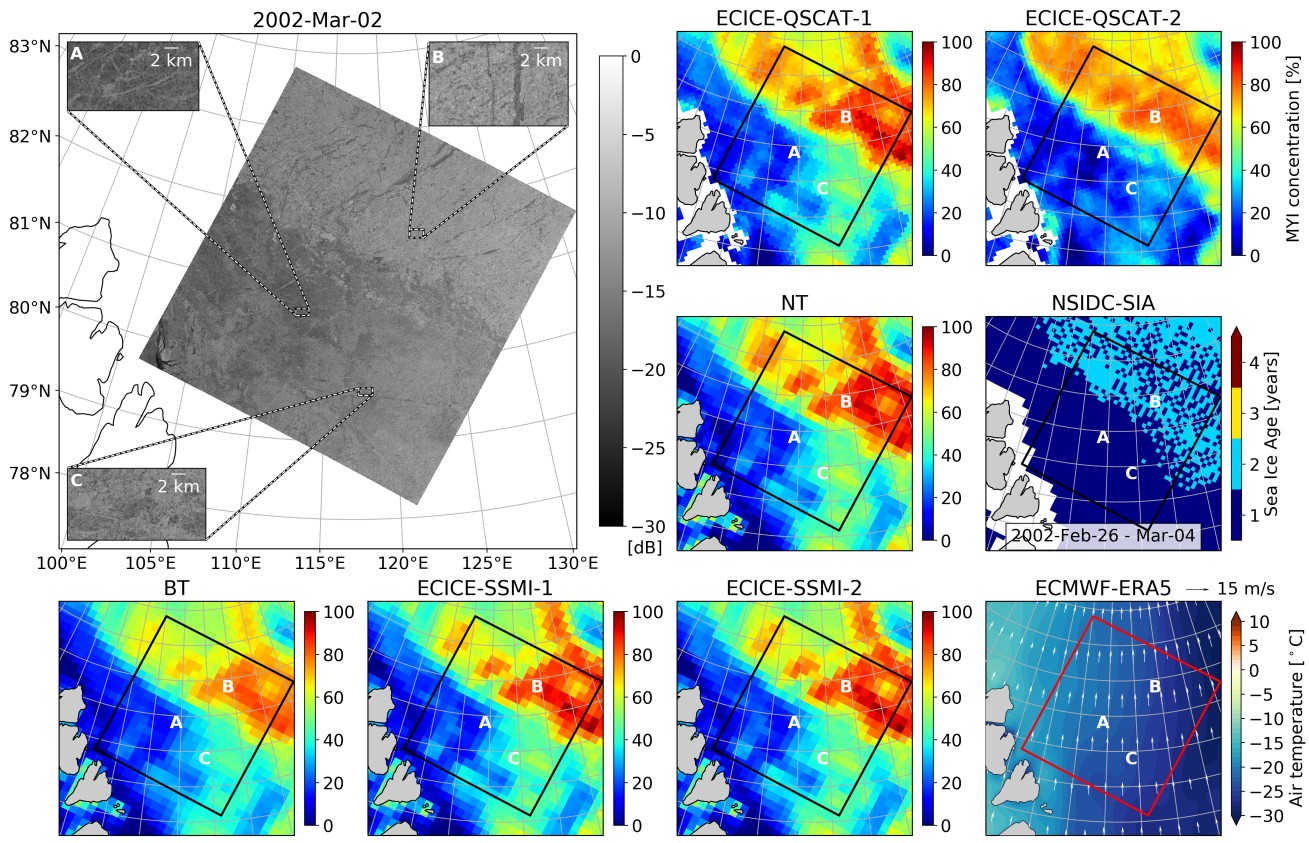

**Figure 10.** RADARSAT-1 image, MYI concentrations from six SITC methods, sea ice age distributions from the NSIDC-SIA product, and 2 m air temperature along with 10 m wind fields from ECMWF ERA5 on March 2, 2002.

A RADARSAT-1 image of a scene in the Laptev Sea, acquired on March 2, 2002 is shown in Figure 10 along with MYI concentration maps from six SITC methods. Enlargements of three areas, labeled A, B and C, are shown in the RADARSAT-1 image and marked correspondingly in other maps. The wind was blowing from the south towards the north with speed around 15 m/s, and the air temperature was between -20 °C and -25 °C. There is a clear boundary separating the northeast and southwest parts in the SAR image. The northeast part is mostly covered by MYI with its high backscatter and visible floe

structure (Area B). Area A features ridges crossing an ice surface with smooth texture and relative low backscatter, which appears to be FYI. Backscatter signature in area C is slightly brighter than that in area A, however its smooth texture makes it more likely to be of FYI. The bright backscatter indicates that it can be rough/deformed FYI. Information from the SAR image agrees well with the sea ice age map, which shows FYI for areas A and C, and SYI in area B. Among all the SITC methods, ECICE-QSCAT-2 agree best with the SAR image. MYI concentration map from the ECICE-QSCAT-2 method shows values

of 80%-90% in area B, and below 30% in areas A and C, while all the pure radiometer-based methods (ECICE-SSMI, NT



and BT) have non-uniformly distributed MYI concentrations between 50%-100% in area B, below 30% in area A and around 60% in area C. Discrepancies in area C indicate that all the purely radiometer-based methods and ECICE-QSCAT-1 tend to misidentify deformed FYI as MYI under high wind conditions.

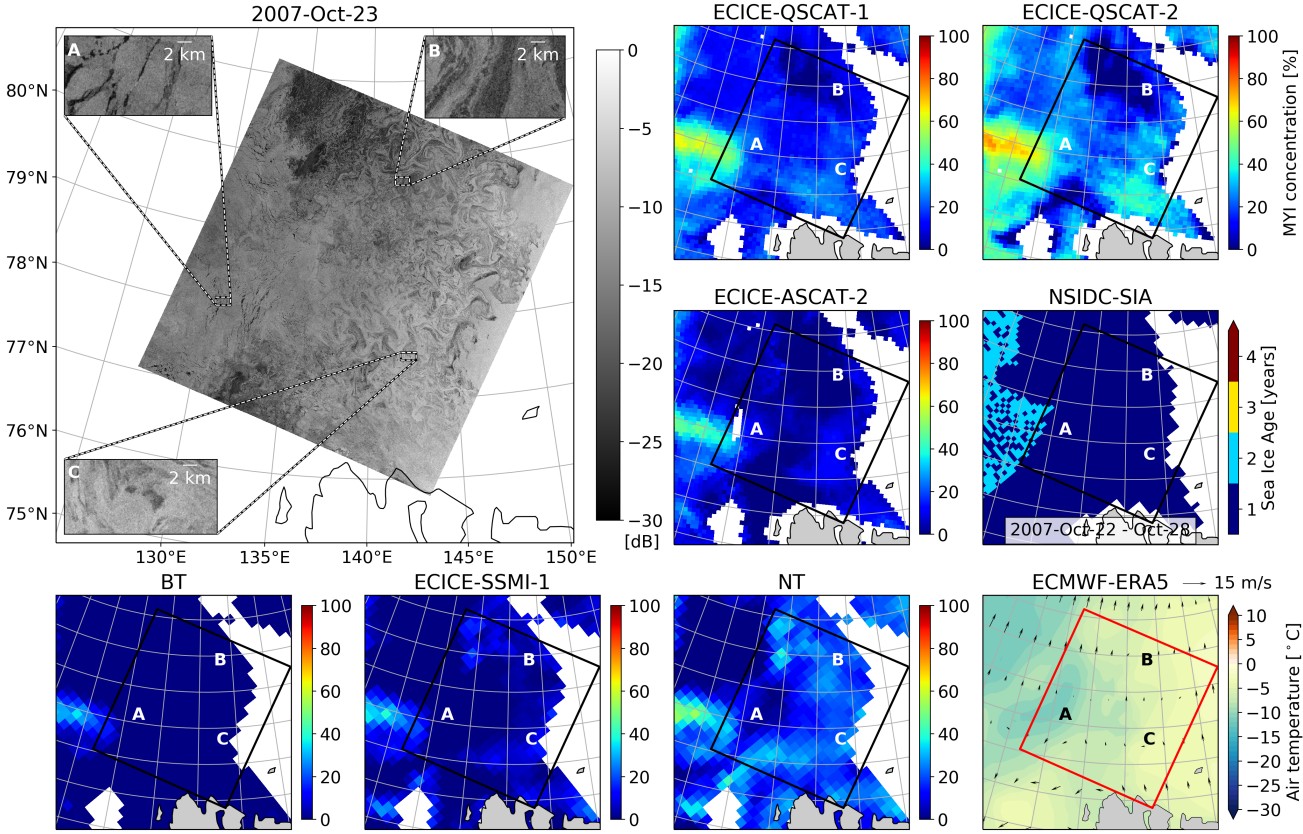

**Figure 11.** RADARSAT-1 image, MYI concentrations from six SITC methods, sea ice age distributions from the NSIDC-SIA product, and 2 m air temperature along with 10 m wind fields from ECMWF ERA5 on October 23, 2007.

Figure 11 shows another case from the Laptev Sea, on October 23, 2007. Area A has typical MYI features in the SAR
image, with high backscatter, texture and ice floe structure. In areas B and C, the RADARSAT-1 image reveals high backscatter with streaks of low values, a typical signature of young ice, which is probably generated due to the low wind and sub-zero temperatures in this area. Backscatter in area B is slightly lower than that in area C, which could be caused by larger fractions of smooth ice or water within a resolution cell. The ECICE-QSCAT-2 method gives fairly good estimates of MYI in areas A and B, however misidentifies FYI as MYI in area C. The ECICE-ASCAT-2 method has relatively good estimates of MYI
in areas B and C, yet it underestimates MYI concentrations in area A. Generally speaking, the ECICE-QSCAT-1 method has equally good performance as ECICE-QSCAT-2 and ECICE-ASCAT-2. On the other hand, the sea ice age map from NSIDC does not show the exact same pattern as the SAR image, since the sea ice in that region could have drifted away after a week





(NSIDC-SIA is a weekly product). In this case, although the use of scatterometer data helps to improve the estimation of MYI concentration, it tends to misidentify young ice as MYI when backscatter from the former is high.

### 4.4.2   Case study (3): ice in the East Siberian Sea

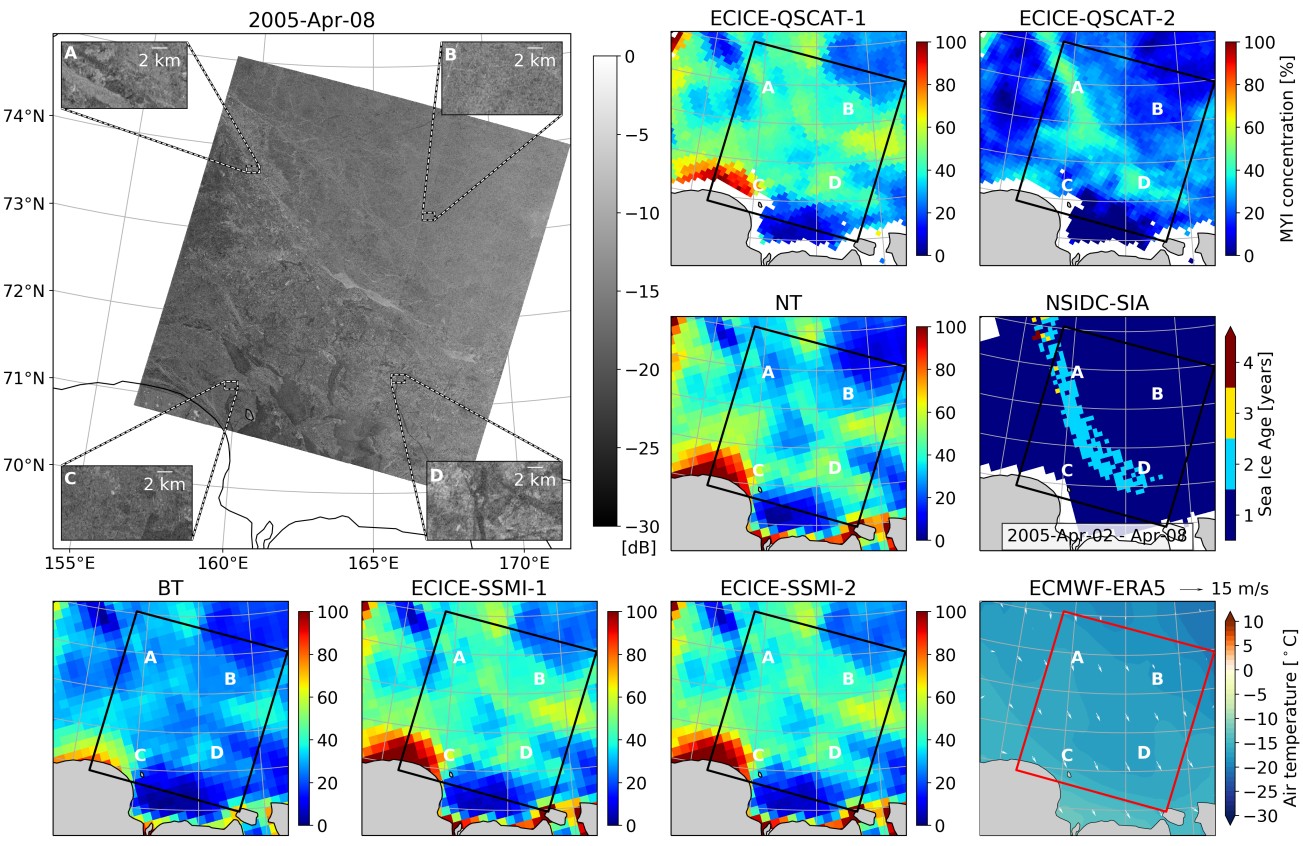

**Figure 12.** RADARSAT-1 image, MYI concentrations from six SITC methods, sea ice age distributions from the NSIDC-SIA product, and 2 m air temperature along with 10 m wind fields from ECMWF ERA5 on April 8, 2005.

The East Siberian Sea is one of the regions that have the most extensive coverage of landfast ice in the Arctic (Dammann et al., 2019). Figure 12 presents a case of this region on April 8, 2005. The air temperatures were around -15 °C, and the wind was pushing ice towards southeast to the Siberian coast. Areas B and C in the RADARSAT-1 image feature FYI with smooth texture and relatively low backscatter, whereas it is shown as FYI in the sea ice age map and low MYI concentrations

only from ECICE-QSCAT-2 (below 40%). Some MYI floes are observed in areas A and D in the SAR image. The elongated feature that goes through areas A and D, south of the bright line in the SAR image, is well preserved in the sea ice age map and ECICE-QSCAT-2 map however not visible for other SITC methods. The SITC methods have an overall overestimation of MYI concentrations except ECICE-QSCAT-2 and BT. While the BT method underestimates MYI concentrations, the good





agreement between the elongated feature with rough texture in the SAR image and the intermediate MYI concentrations in

ECICE-QSCAT-2 serves as another proof for the good performance of ECICE-QSCAT-2.

### 4.4.3   Case study (4): ice in the Beaufort Sea

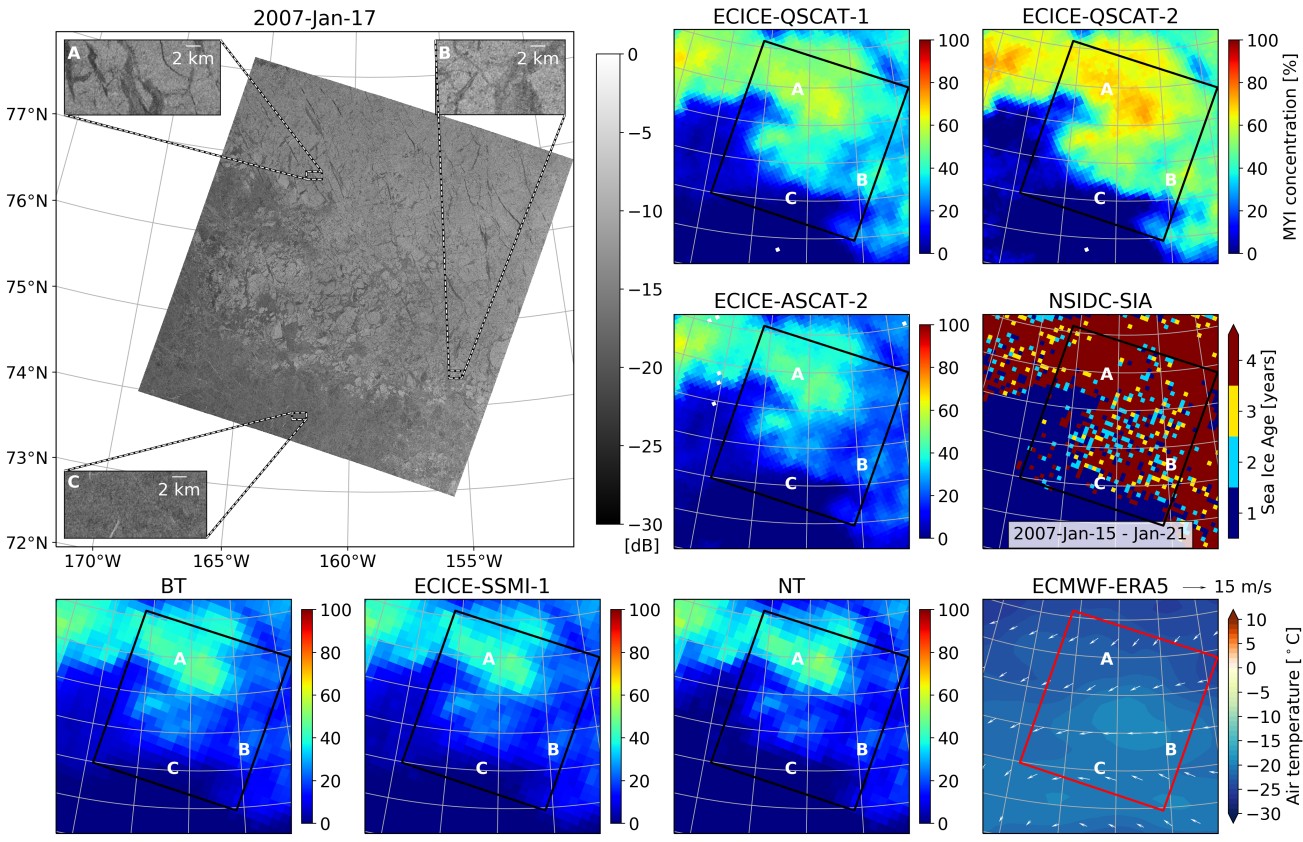

**Figure 13.** RADARSAT-1 image, MYI concentrations from six SITC methods, sea ice age distributions from the NSIDC-SIA product, and 2 m air temperature along with 10 m wind fields from ECMWF ERA5 on January 17, 2007.

Figure 13 presents a case from the Beaufort Sea. The Beaufort Sea is a highly dynamic ice regime, where the ice motion is characterized by large-scale anticyclonic circulation known as the Beaufort Gyre. On 17 January 2007, the air temperature was between -10 °C and -15 °C, while the wind was blowing from the north and east to the southwest. There is a clear distinction

between area A/B and area C. Areas A and B contain MYI identified in SAR image by its high backscatter and visible floe structure, whereas area C feature ice surface with low backscatter and smooth texture, which are typical features of FYI. Among all the SITC methods, MYI concentrations from ECICE-QSCAT-2 have the best agreement with these features, with values of 70%-80% in area A, 40%-60% in area B and below 15% in area C. ECICE-QSCAT-1 and ECICE-ASCAT-2 have the second and third best performance. This case shows that using scatterometer data in SITC methods favors the identification of MYI in





highly dynamic regions. Moreover, the performance is better with QSCAT (Ku-band) than ASCAT (C-band). As it is shown in Rivas et al. (2018), the separability between deformed FYI and MYI is better at Ku-band than C-band. Explanations can be found in a study by Ezraty and Cavanié (1999). Ku-band and C-band are similarly responsive to surface roughness, e.g., over deformed FYI, but Ku-band, with its wavelength around 2.2 cm, matches the characteristic dimension of air bubbles in MYI better.

**4.4.4  Case study (5): ice around the Banks Island**

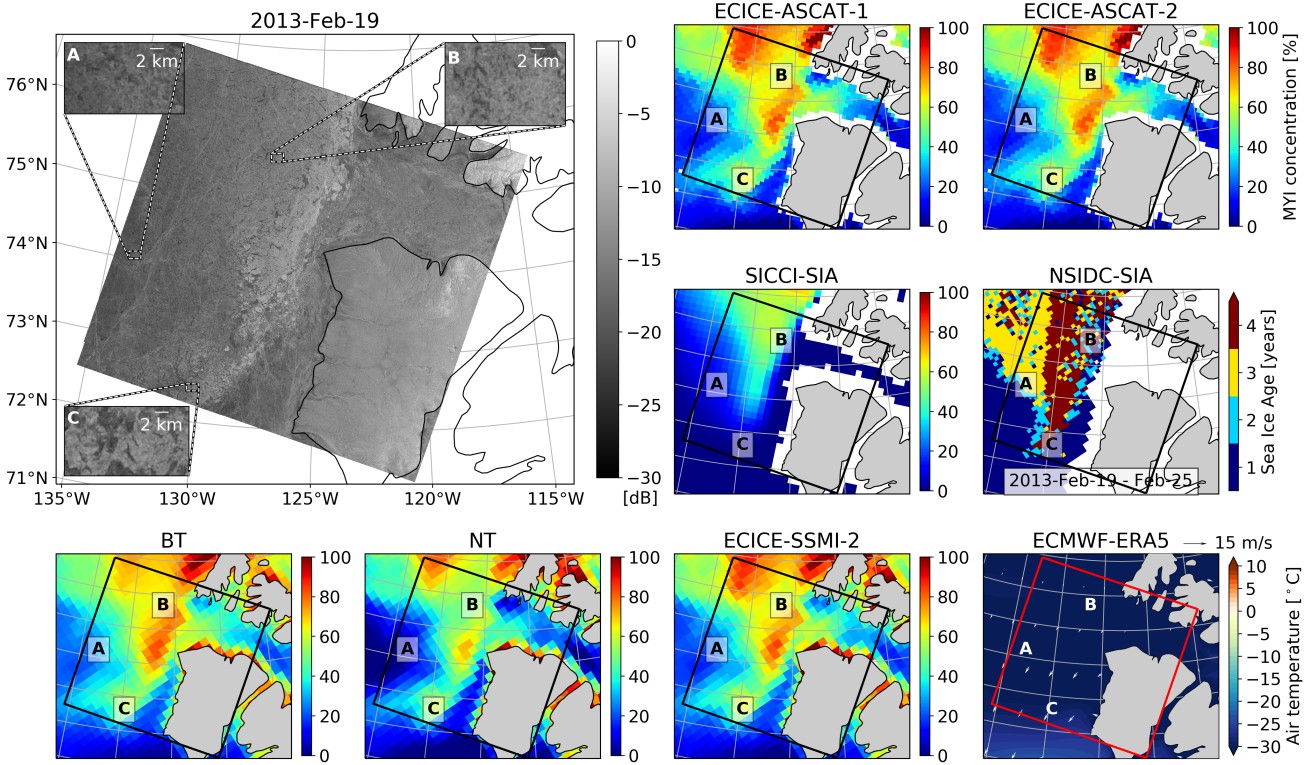

**Figure 14.** RADARSAT-1 image, MYI concentrations from six SITC methods, sea ice age distributions from the NSIDC-SIA product, and 2 m air temperature along with 10 m wind fields from ECMWF ERA5 on February 19, 2013.

Figure 14 presents a case around the Banks Island in the Beaufort Sea on February 19, 2013. The three enlarged areas (A, B and C) have fairly similar backscatter signatures in the RADARSAT-2 image, especially areas B and C, both with clear ice floe structure and bright backscatter. In area A, the backscatter is not as high as that in areas B and C, however its texture and floe-like structure make it appear to be partly covered with MYI. In this case, MYI concentrations from all the SITC methods

have similar distribution pattern, with values between 50% and 90% in areas B and C, and below 40% in area A. In comparison, the two SIA products give slightly different estimates than the SITC methods and SAR image. NSIDC-SIA misidentifies MYI as FYI in the southwest part of the SAR image (west of area C), whereas SICCI-SIA underestimates MYI concentrations in





area C. Mismatches between the NSIDC-SIA product and SAR image can be partly explained by the low temporal resolution (weekly) of NSIDC-SIA, while the main reason is probably the incorrect representation of ice drift, which is the basis for all

SIA products and is difficult to calculate when there is no distinct characteristics to track. As reported in Korosov et al. (2018), the overall homogeneous distribution of MYI contration in SICCI-SIA is due to the void of potential artifacts in the ice drift product that is used for sea ice age tracking. On the other hand, the sea ice concentration product used in SICCI-SIA leads to an overall underestimation of old ice fraction.

### 4.4.5  Case study (6): ice in the Fram Strait

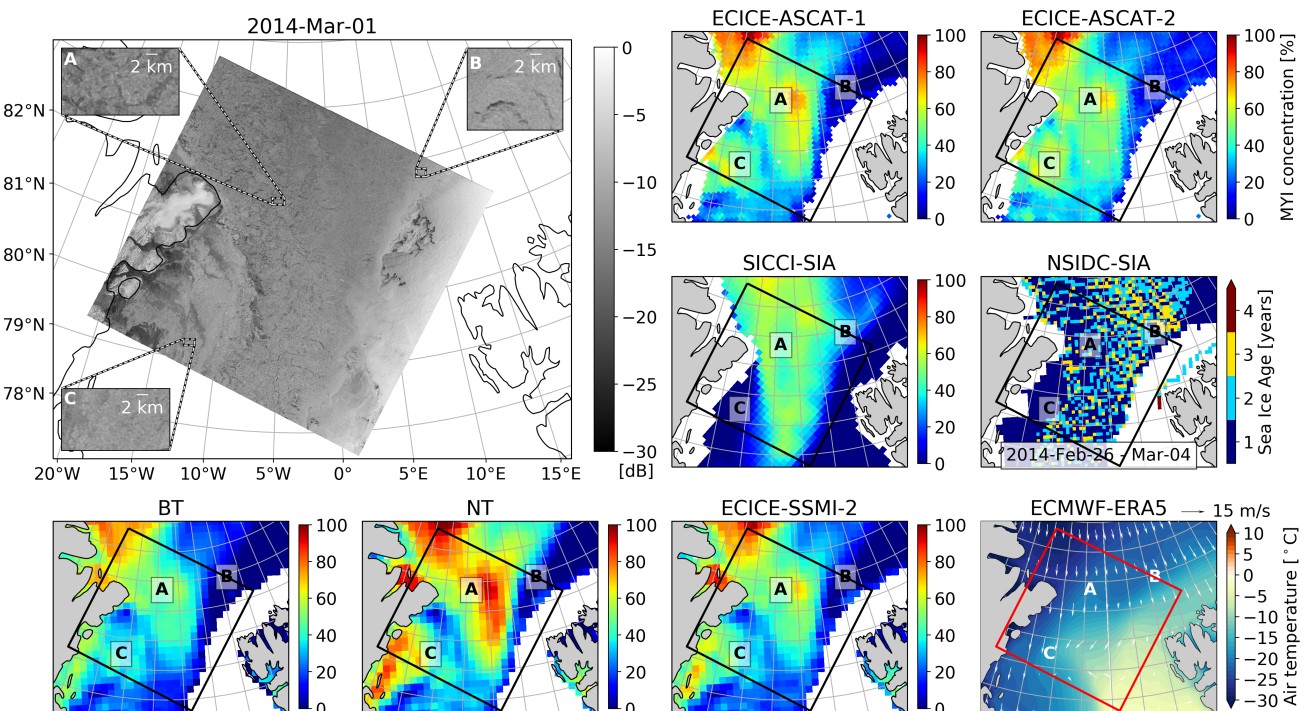

**Figure 15.** RADARSAT-1 image, MYI concentrations from six SITC methods, sea ice age distributions from the NSIDC-SIA product, and 2 m air temperature along with 10 m wind fields from ECMWF ERA5 on March 1, 2014.

A case from the Fram Strait is shown in Figure 15. The Fram Strait is the main gate of Arctic sea ice outflow (including MYI). It features high ice concentrations and large ice motions. On March 1, 2014, the temperature was decreasing from the southeast to the west, with values between -5 °C and -15 °C, and the wind was blowing from the north and northeast to the south. Area A has typical MYI features in the SAR image, with rough texture and clear ice floe structure. All the SITC methods are able to identify some MYI though with different distributions of concentrations. In area B, the SAR image reveals linear patches

of low backscatters surrounded by high backscatters with relative smooth texture. Given the particular location, between the pack MYI and open ocean, and the high wind speed, it is quite likely to be brash ice in area B, which is typical in the highly





dynamic marginal ice zone. In this area, all the SITC methods manage to give near-zero MYI concentrations, while the two SIA products both misidentify FYI as MYI. Another discrepancy between the SITC and SIA results is for area C, where the SAR image reveals bright backscatter signature yet smooth texture, which appears to be deformed FYI. In contrary to the situation

in area B, the SIA products are able to identify FYI but not the SITC methods. This case, along with the previous one, proves that sometimes the SITC methods can work better than the SIA products on ice type discriminations, and in other cases the SIA products capture better the ice type distributions than most of the SITC methods. It is important to have both products for sea ice type distribution monitoring.

## 5   Conclusions

Results from the eight SITC methods are inter-compared with two SIA products and three SIT products, using microwave radiometer and scatterometer data spanning the winters (October - April) from 2000 to 2015. Performances of the SITC methods are evaluated quantitatively and qualitatively with samples that are used to generate tie points as well as SAR images. Overall, the results can be summarized as follows:

- Using scatterometer data along with radiometer data helps the discrimination of FYI and MYI. The most appropriate
approach to reproduce sea ice type distributions in the Arctic is the combined method: ECICE-QSCAT-2 for the years 2000-2009 and ECICE-ASCAT-2 (or ECICE-ASCAT-1) for 2009-2015.

- In the SITC methods, Ku-band scatterometer works better than C-band scatterometer on identifying MYI due to the higher sensitivity to volume scattering in MYI. On the other hand, performance of the QSCAT-based methods depends on the input parameters (combination of observations from scatterometer and radiometer), whether they contribute to the
distinction of ice types under different conditions.

- Among the pure radiometer-based SITC methods, the ECICE-SSMI methods work slightly better than the NT method on identifying MYI, while the BT method has an overall underestimation of the MYI concentration.

- The NSIDC-SIA and SICCI-SIA products work fairly well on identifying the ice types that can be distinguished from the eight SITC methods, however they highly rely on the ice drift product used in the algorithm. The weekly NSIDC-SIA
product does not provide ice age information as precisely as the SICCI-SIA product, due to the lower temporal resolution and its monitoring of oldest ice. The SICCI-SIA product, on the other hand, seems to underestimate MYI concentration in homogeneous regions with little artifacts to track.

- Among the three SIT products, the eight SITC methods have the best agreement with the KNMI-QSCAT-SIT product. Besides, all the SITC methods agree better with the SIT products in mid-winter than transition seasons.

- Although the SIA and SIT products are fairly good datasets for delineating ice type distributions, the SITC methods are better on preserving details like varied concentration of different ice types and work better under specific sea ice conditions.





Among the eight SITC methods, the most suitable datasets for climate studies are the retrievals from ECICE-QSCAT-2 and ECICE-ASCAT. However, even for the most appropriate datasets, there are still problems of overestimation and underesti-
mation that are impacted by meteorological events such as warm/cold air spells or wind-driven factors like ice deformation. Corrections that use information of air temperature and ice motion (Ye et al., 2016b, a) could be used to mitigate these inevitable problems. Apart from being a good dataset for monitoring MYI coverage changes, the corrected MYI concentration data can also be used for defining ice and snow densities for ice thickness retrieval algorithms, ice roughness for the ice circulation models and so on. Moreover, the new Ku-band scatterometer on China-France Oceanography SATellite (CFOSAT)
opens another opportunity of having better MYI dataset for polar and global climate monitoring. In recent comparative studies of many sea ice concentration retrieval algorithms (Ivanova et al., 2014; Lavergne et al., 2019), it is shown that using lower frequency microwave observations at 6.9 GHz are less influenced by the atmosphere so that algorithms based on them yield sea ice concentration with better accuracy. However, because of the low horizontal resolution at 6.9 GHz (e.g. from AMSR-E/2 with 50 km), such algorithms are not applied operationally. This situation may change with the planned radiometer Copernicus
Imaging Microwave Radiometer (CIMR) with an antenna reflector of about 8 m diameter and resolution at 6.9 GHz of 15 km on surface (Kilic et al., 2018). We expect that then, also SITC can be estimated at a better accuracy. For the time being, the AMSR-E/2 data available since 2002 may be used as model data for developing a 6.9 GHz SITC retrieval algorithm.

*Data availability.* Numerous data sets used in this study are publicly available and were obtained from the following locations:

QSCAT data from CERSAT/IFremer, http://products.cersat.fr/details/?id=CER_PSI_ARC_1D_012_PSI_QS (last access: 28 August 2019);
ASCAT data from CERSAT/Ifremer, http://products.cersat.fr/details/?id=CER_PSI_ARC_1D_012_PSI_ASCAT (last access: 28 August 2019);

SICCI-SIA product, ftp://ftp.nersc.no/ArcticData/esa_cci_sea_ice_age/ (last access: 28 August 2019);

NSIDC-SIA product, https://nsidc.org/data/nsidc-0611 (last access: 28 August 2019);

C3S-SIT product, https://cds.climate.copernicus.eu/cdsapp#!/dataset/satellite-sea-ice?tab=overview (last access: 28 August 2019);
KNMI-SIT product, http://projects.knmi.nl/scatterometer/ice_extents/ (last access: 28 August 2019);

ECMWF ERA5 data, https://cds.climate.copernicus.eu/cdsapp#!/dataset/reanalysis-era5-single-levels?tab=overview (last access: 28 August 2019).

*Author contributions.* Y.Y. conducted the data analysis and lead the manuscript writing. M.S. provided access to the SAR images and thorough review. S.A. provided the gridded radiometer data and contributed to the research design. W.A. and L.E. contributed to interpretation
of the SAR images. G.H., C.M. and F.G. contributed to the research design. All co-authors participated in fruitful discussions and writing of the manuscript.

*Competing interests.* The authors declare that they have no conflict of interest.



*Acknowledgements.* This work was supported by the Area of Advance at Chalmers University of Technology, Gothenburg, Sweden. We thank the ASF and MDA for providing RADARSAT-1 and RADARSAT-2 images. The contribution of C.M. was supported by the Deutsche

Forschungsgemeinschaft (DFG) in the framework of the priority programme "Antarctic Research with comparative investigations in Arctic ice areas" by grant SP1128/2-1,



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
