# Peer review of "Inter-comparison and evaluation of sea ice type concentration algorithms"

_The Cryosphere, 2019_

## Referee Comment (RC1) · Anonymous Referee #1 · 9 Dec 2019

General comments:

There are a number of flaws in this study, with inconsistencies between its objectives, the way it is conducted, and its conclusions.

Climate monitoring is clearly specified as the objective (l. 39 l. 388), but the method finally suggested cannot be used for long climate record. The data are only available starting in 2000. If the final goal is to produce long climate record, the selected method has to be applicable over long time series, otherwise the objective of the study is not met. The interest of this method with respect to the ones that are applicable to longer time series is not clear and needs to be discussed further.

The comparisons are only applied to winter months. Why not summer months as well?

[Figure]

The reviewer is well aware that the algorithms have strong limitations during summer months. However, the real problems have to be tackled. From a year to the next, the sea ice decline essentially occurs during the summer months, not during the winter months. Would it be interesting as well to evaluate the methods during that period? It is necessary here to better justify the use of winter data only, if summer data are not explored.

Sea ice types are FYI or MYI. The type is thus directly related to the age of the ice. The paper uses two sea ice type algorithms. It also describes and uses sea ice age algorithms. From a naïve point of view, the two variables should be related, with the sea ice type derived from the sea ice age. Can you elaborate on the fundamental difference between the nature of the two variables and their relative merits? It is actually very surprising to see that the algorithms for sea ice types and sea ice ages are generated by partly common producers, but with very different methods for the two products and consequently possible inconsistencies. Consequently, the first step is to compare the two types of products (sea ice type and sea ice age), and if not consistent, they should be reconciled (especially if they are generated by the same groups).

Snow impact on the signal is mentioned (l. 33), but not discussed in the rest of the paper.

The study should focus on the analysis of a limited number of carefully selected products, based on the underlying physics of the observations involved in the algorithms, and on the method used in the algorithms, having in mind the improvement of consistent climate series.

More specific comments:

The title of Section 3.1 mentions three sea ice type concentration algorithms. Finally, eight of them are compared, six of them from the same provider, and with common features between them and with the two others. Four are only derived from passive observations, with a maximum of four channels (19 and 37 GHz, possibly dual polarizations) at 25 km resolution. The four others are also using active microwaves, and they benefit from a twice better spatial resolution. Note that it is not clear if it is the spatial resolution of the product that is mentioned or its sampling. Any possibility to limit the number of products for comparisons, based on the physics of the observations? From the six products provided by ECICE, which ones are recommended by the provider? I would understand the comparisons of products with rather different passive microwave observations, with lower frequencies for instance. But here it is just the comparison of different ways of combining passive microwaves that have been around for several decades (starting in the 80s). In addition, the fundamental differences between the C and Ku bands of ASCAT and QSCAT is not discussed much, as well as their differences in terms of incidence angles or polarizations. These are real issues when building climate series. Similar comments apply to the sea ice age algorithms and sea ice type algorithms, with a lack of explanation of their differences in terms of physics.

The tie points (section 3.2) are estimated on data covering almost all year. Why not extending them to the year for the derivation of a MYI product applicable to the full year? Would it be possible to analyze the seasonality of the responses on Figure 3? That could justify the use of winter months only. It would be also very informative to present similar results for the ASCAT and QSCQT backscattering parameters (same as Figure 3, with addition of information on the seasonal cycle).

Figure 4 raises some questions. It seems that, regardless of the SIA product (SICCI or NSIDC), the extent of the multi-year ice steadily declines during the winter (ice with age > 1yr). I would expect it to decrease during the summer, but not during the winter months. If one makes the fair assumption that the sea ice does not melt during the first part of the winter, it means that the multi-year ice is transformed in FYI during the winter? That would be an interesting rejuvenating process! In the ECICE products, this is not observed: in general, MYI is stable or slightly increases during the first part of the winter, as expectation. The behavior of the SIAs seems very questionable and

should be commented. Still on Figure 4, from April 2016 to October 2016, the MYI extent from SICCI-SIA is constant: this should mean that all the FYI melted during that summer. Is that confirmed by other observations? Note that the NSIDC-SIA product does not observe it and there is a ratio of $\sim$1.6 between the two SIA estimates for that month, raising doubts again on the SIA products. These key aspects have to be clarified before any quantitative analysis of the differences between the ECICE and the SIA in 4.1.1.

As already said, comparison between the SIT and SIA have to be performed before other comparisons, as these products are both related to the ice age. Figures 6-7 actually provide an indirect comparison of the two SIT products. It would be much more efficient and clearer to directly compare the two SITs. The conclusion from the comparisons between the SIT and the SITC that the ECICE QSCAT are the best SITC product is very surprising and not convincing (l. 244 and following). The 'validation' with the tie point is not more convincing. The ECICE QSCAT 1 and QSCAT 2 products provide rather different results, especially at the beginning of the winter season (Figures 6-8-9). If it was the addition of the QSCAT that was beneficial to the retrieval, more stable results would be expected between QSCAT 1 and QSCAT 2. The spatial resolution issue is never discussed in the analysis, but all the ECICE products with QCSQT and ASCAT have twice the spatial resolution of the purely passive microwave products. That can clearly help the comparison with other products. Second the ECICE QSCAT 2 is the only product without the 19 GHz channel (19GHz has twice worse spatial resolution than the 37 GHz channels). That can help improve the final spatial resolution of the product for comparison with other parameters. All these factors can play a role in the differences between the algorithms, and should be discussed. Note that the notion of spatial resolution in this paper is very unclear, as it is likely closer to the spatial sampling of the data.

The validation with the SAR data could be very interesting. However, it is not quantified and only based on the qualitative examination of SAR images, without much information about the methodology. It is rather difficult to evaluate the different SITC and SIA products from these qualitative comments. The incidence angles of the SAR observations have to be mentioned. Does it change much from one side of the image to the other? No SIA product is shown in these comparisons. Why? Seems like there is a systematic problem in the color bar of the SICCI-SIA problem (not age but percentage). The SAR and the scatterometer data are based on the same physics (backscattering from a surface), and ASCAT and the SAR share the same frequency. It is expected that the SAR and the scatterometer images show some similarities, at least qualitatively and at large scale. These comparisons between the SAR data should again concentrate on a few carefully selected parameters, with discussion on the underlying physics and algorithm principles.

If used in climate data records, the transition from the ECICE-QSCAT and the ECICE-ASCAT algorithms have not been analyzed. Is it smooth enough for climate studies? Jumping from C to Ku band is very likely problematic. In addition, nothing is said about the possible use of ERS data before, to extend the data set.

---

## Referee Comment (RC2) · Anonymous Referee #2 · 20 Dec 2019

Summary:

This paper compares eight different sea ice type concentration products, consisting of passive microwave derived products and combined scatterometer/passive microwave products. Evaluation is done in comparison to three sea ice type products and two sea ice age products. Quantitative analysis is done via comparison with tie point values (pure surface types) and qualitative analysis done via comparison with interpreted SAR imagery. The results indicate that the best overall SITC performance is from the combined scatterometer and passive microwave products.

General Comment:

This paper provides a quite thorough assessment of several different SITC products

and the conclusions are well-supported. It makes sense that using the combination of passive and active sensors would perform best. The analysis demonstrates the differences between the various products quite well and thus acts as a solid baseline reference point for understanding uncertainties in the different products. SITC is an important parameter, increasingly so as the Arctic is transitioning from primarily MYI-dominated regime to one that is FYI-dominated. While SIT (MYI vs FYI tagging of pixels) is fairly common, useful SITC is more challenging. This paper shows the value of the SITC products and indicates the best products. The paper is well-written, the results are logically presented, and figures are for the most part clear (comments below) and illuminate the results. I recommend publication after address a few minor issues.

One overarching thing is that the paper does not address summer melt at all. I know that summer melt obscures the backscatter and emission from the ice and SITC are not retrievable. But a reader not as familiar with microwave characteristics of sea ice may not realize that. Nowhere in the paper is the melt effect mentioned. I think a sentence or two in a logical place early in the paper (e.g., somewhere in the Intro or Section 2.1) to note this and explain why your results only cover the months of October through April and why all the case study comparisons with SAR take place during those months. Other comments are below by line number.

Specific Comments:

80: I'm a little confused by the use of "backtracking". At least for the NSIDC-SIA, I would simply call it "tracking". I guess it's backtracking in the sense of one is counting from back when ice first formed. But the actually method simply tracks parcels in ice forward in time an increments the age.

133: "OW" is used here without explanation. It should be spelled out as "Open Water" the first time it is used.

139, Table 1: It's good to note the resolution in the table, but it seems like the differences in the resolution might play some role? The products that include scatterometer

data perform the best, but these are also the products and double the spatial resolution. So, there is the question of whether the scatterometer input is what performs better or if it is the better resolution? I actually think it is largely the inclusion of backscatter as another source of information, with better resolution probably being fairly minimal. However, better resolution may have some effect and I think this is worth acknowledging as a potential (small) contributor.

157: This a pet peeve of mine, but I don't like the use of "proof" in science papers. Science is always provisions, subject to further data or evidence. In particularly "concrete proof" here is much stronger than warranted. I would suggest "concrete evidence" or something similar. This occurs elsewhere (as I point out below, but I may have missed some).

179: "It can be seen. . ." add "in Figures 4 and 5"

181: Typo "NSIDC"

218: "sets of ice pixels" sounds odd to me. Maybe "regions of interest"?

Figures 6-8: These are nice and I like how you see the year-to-year variation. But it does crowd things. I would change these figures, but I might suggest adding a figure (or figures) that show an average seasonal cycle from all the years. Then you could have a plot that is just October-April and it would be easier to see the differences between the products. I realize that characteristics vary between years, but still I think an average seasonal cycle might make it easier to distinguish the major features of the various products.

Table 4: What do the bolded numbers indicate?

Figure 9: I wonder if maybe instead of grouping the 8 products by 5% bins, group the concentrations by the product. Either way there are 8 categories. But I think having all of the % bins together for each product would be clearer. I find myself trying to read the different % bins of each product and that's difficult to do as they are plotted. I might

also be careful about the colors – there may well be color blindness issues with the color palette used.

269: Another "proof", suggest "evidence" instead.

275ff: This is really nicely done – the figures look great and overall everything is quite clear. The main question here is who is interpreting the SAR imagery and how? SAR sea ice imagery can be quite difficult to interpret in some cases and the backscatter characteristics can vary for different ice types. This is why usually interpretation is done at ice charting centers by expert trained analysts. I don't disagree with the interpretation, but I'm not an expert in SAR sea ice imagery. So, I just wonder who interpreted the images and what is the basis for their interpretation. There are sometimes "analyzed" SAR imagery done at ice charting centers – i.e., imagery with outlines of different ice types drawn in by analysts. This would be ideal, but I don't think such annotated imagery is generally available. Again, I think the approach is fine and the interpretation seems reasonable, but I'd like a little more detail on the basis for the interpretation. Otherwise, it sounds a bit "hand-wavy".

Figures 10-15: As note, overall these figures are very nice. The one thing a little hard to discern is the SAR images. Is there a way to increase the contrast on the images? It's hard to distinguish some of the subtleties in the grey scale for some of the regions discussed. The other thing a little hard to see is the wind vectors. They're quite small and thin.

275ff: As noted above, the fact that retrievals are not possible during summer is not stated in the paper. And Figures 10-15 are between October and April to avoid the melt issue. One question is how "close" to the "shoulder" season can one go? I guess once melt appears in the signal, you lose ice type information. But the exact timing depends on location and local conditions. For example, the Fram Strait example, on March 1, seems like it could (if conditions were right) cause surface melt, though in this case the winds are from the north. But temperatures do appear to be approaching 0 C near the
ice edge. This comment is mainly just to further suggest that melt effects should be discussed and this section may be one area to include that.

319: "proof"

360: "proves"

376, 378: I prefer "perform" over "work", which sounds colloquial to me.

382: "few" instead of "little"

---

## Author Comment (AC1) · 11 Feb 2020

Responses to the comments, revised manuscript and marked track changes are attached as tc-2019-200-AC1-supplement.zip file.

Please also note the supplement to this comment:
https://www.the-cryosphere-discuss.net/tc-2019-200/tc-2019-200-AC1-supplement.zip

---

## Author Comment (AC2) · 11 Feb 2020

Responses to the comments, revised manuscript and marked track changes are attached as tc-2019-200-AC2-supplement.zip file.

Please also note the supplement to this comment:
https://www.the-cryosphere-discuss.net/tc-2019-200/tc-2019-200-AC2-supplement.zip